# Monkeypox Cross-Sectional Survey of Knowledge, Attitudes, Practices, and Willingness to Vaccinate among University Students in Pakistan

**DOI:** 10.3390/vaccines11010097

**Published:** 2022-12-31

**Authors:** Narendar Kumar, Fatima Ahmed, Muhammad Sauban Raza, Pushp Lata Rajpoot, Wajiha Rehman, Shoaib Alam Khatri, Mustapha Mohammed, Shaib Muhammad, Rabbiya Ahmad

**Affiliations:** 1Discipline of Clinical Pharmacy, School of Pharmaceutical Sciences, Universiti Sains Malaysia, Penang 11800, Malaysia; 2Department of Pharmacy Practice, Faculty of Pharmacy, University of Sindh, Jamshoro 76060, Pakistan; 3Rural Health Centre Headrajkan, Tehsil Yazman, Bahawalpur 58240, Pakistan; 4Rural Health Centre Gogran, Lodhran 58240, Pakistan; 5Department of Health Education and Promotion, College of Public Health and Tropical Medicine, Jazan University, Jazan 45142, Saudi Arabia; 6Department of Health Informatics, Faculty of Public Health and Tropical Medicine, Jazan University, Jazan 45142, Saudi Arabia; 7Sindh Government Hospital, Korangi No. 5, Karachi 74000, Pakistan; 8Department of Clinical Pharmacy and Pharmacy Practice, Faculty of Pharmaceutical Sciences, Ahmadu Bello University, Zaria 810282, Nigeria

**Keywords:** knowledge, attitude, perception, monkeypox virus, smallpox vaccine, Pakistan

## Abstract

This study aimed to explore knowledge, attitude, perceptions, and willingness regarding vaccination among university students in Pakistan. This cross-sectional study was carried out using an open online self-administered survey via Google Forms. The survey data were collected between the 15 to 30 of October 2022. A total of 946 respondents participated in the study, of which the majority were female (514, 54.3%). Most students belonged to a medical background, specifically pharmaceutical sciences. Most of the respondents did not know about monkeypox before 2022 (646, 68.3%). Regarding overall knowledge of monkeypox, most of the respondents had average knowledge (726, 76.7%), with very few having good knowledge (60, 6.3%). Regarding overall attitudes towards monkeypox, most of the respondents had neutral attitudes (648, 68.5%). There was a significant association between knowledge of Monkeypox with the type of academic degree (*p* < 0.001), type of discipline (*p* < 0.001), and region of respondents (*p* < 0.001). The willingness to vaccinate among the population was (67.7%). The current study pointed out that the overall knowledge of monkeypox was average in most respondents, with considerable knowledge gaps in most aspects. The overall attitude towards monkeypox was neutral. Further, the knowledge about monkeypox was strongly associated with academic degree, study discipline, and region of respondents. Our findings emphasize the need to raise public awareness by educating students on the monkeypox virus. This will improve adherence to preventative recommendations.

## 1. Introduction

A zoonotic disease caused by a monkeypox (Mpox) virus infection is becoming a new concerning disease and is known as Mpox. The virus belongs to the Orthopoxviral genus [1]. The very first case, which became the basis for the endemic nature of the virus, was reported in the Democratic Republic of Congo (DRC) in the year 1971. After that, the infection spread as endemic in central and West Africa [2,3]. Then in 2003, due to rodents imported from Ghana, the first case in the US was reported. [4]. On 23 July 2022, the disease was declared a public health emergency of international concern by the Director General of WHO as Mpox outbreaks in many countries [5]. As of 27 November 2022, more than 81,000 cases and 55 fatalities have been reported in 110 countries around the globe [6].

The clinical manifestations of Mpox are similar to that of smallpox, such as pyrexia along with pain in the back and head, rash, malaise, and fatigue, except for a particular symptom, i.e., lymphadenopathy [7,8]. The virus is transmitted through bushmeat handling, animal hosts such as squirrels, rodents, and prairie dogs, infected oropharyngeal secretion, and direct contact [9]. In addition, various risk factors, such as human interaction with infected animals, smallpox vaccination termination, and increasing globalization, mean Mpox is a future public health concern on a global level [10].

Although the mortality rate is low with this virus, it may lead to critical complications [11]. The question of its re-emergence is still unanswered, but the reason behind a potential alarming situation in the future is because of the wide range of animal hosts and its high power of adaptation [12]. One WHO report revealed that the reason behind its re-emergence is a lack of awareness about Mpox. When a study was carried out with the aim of understanding how many general practitioners have sound knowledge regarding Mpox, it was revealed that only 27% (Italy), and 18.6% (Saudi Arabia). Another study carried out in Saudi Arabia revealed that 48% of the general population had sufficient knowledge about Mpox [13,14].

One of the most significant global accomplishments was the eradication of smallpox, which was accomplished through an effective vaccination program. Almost all children and most of the world’s population have little to no protection against orthopoxviruses. Most people are vulnerable to the current Mpox virus threat. Considering the escalating number of Mpox infection cases worldwide, the Advisory Committee on Immunization Practices (ACIP) recommends pre-exposure prophylaxis for health workers, laboratory personnel, clinical laboratory staff, and others who may be at risk of contracting Mpox [15,16].

Pakistan is a low-middle-income country, with inequitable distribution of scarce resources. In the year 2021, the government spent 1.2 percent of its GDP on health; this amount is far less than the WHO recommendation of 5% [17]. The total literacy rate of Pakistan is less than 40% [18]. The number of students who obtain a university education is less than 30%. Due to unawareness and illiteracy, there is a lack of basic understanding of basic health rights. If any pandemic strikes, university students should be more capable of spreading awareness among the masses. This is the main reason to target this population i.e., university students.

The responsiveness of the health system is another major issue, added to by a reactive instead of a proactive approach, i.e., we usually identify problems when they are already complicated. A struggling healthcare system will be on the verge of collapse if monkeypox starts to spread. Pakistan does not have any diagnostic facility for the virus, but the health department has declared that samples can be sent abroad for testing in case of emergencies, which further threatens the spread. To tackle this situation, there is a need to have adequate knowledge regarding the presenting signs and symptoms of the disease to assure the timely quarantine of suspected patients instead of symptomatic treatments only. Additionally, hospitals should be prepared with well-equipped isolation units to quarantine patients immediately to limit the spread of the contagious virus.

The spread of Mpox can be controlled if the general public is educated on the disease and on how to protect themselves from the virus. The objective of the proposed study is to measure the level of knowledge, attitudes, and perceptions about Mpox among university graduates in Pakistan to provide baseline information and comprehension of the necessary subsequent steps.

## 2. Materials and Methods

### 2.1. Study Design

The present study is cross-sectional in nature and was carried out through an open online self-administered survey designed using Google Forms. The survey data were collected between the 15 to 30 of October 2022. The survey was conducted and reported based on the Strengthening the Reporting of Observational Studies in Epidemiology (STROBE) guidelines.

### 2.2. Study Setting and Population

The study was conducted among university students in Pakistan. Participants were drawn from different disciplines, including health-related and non-health-related sciences. Only participants aged 18 years and above, who could understand English, provided informed consent, and were willing to participate voluntarily were included.

### 2.3. Study Instrument

The research instrument was adapted from previous studies [19,20,21]. The questionnaire was drafted in English and designed using Google Forms. Information regarding research background, confidentiality statement, and voluntary participation was included. The survey consisted of four sections: (1) socio-demographic characteristics (gender, age, level of education, discipline, region of residence, and monthly family income), (2) knowledge-based questions, (3) attitude-based questions, and (4) perception-based questions toward Mpox. Additionally, there were four categorical questions regarding previous knowledge about Mpox, whether they had received any information on Mpox in university, and the history of COVID-19 and flu vaccination. All questions were close-ended with categorical options except age (years).

### 2.4. Data Collection

A pilot study was conducted involving 40 students for content reliability. Cronbach’s alpha coefficient was used to measure the internal reliability of the pilot study. The value of Cronbach’s alpha for the survey was found to be 0.78. The survey hyperlink was shared through WhatsApp messenger to invite participants to the study. To check the response, reminders were occasionally sent to the participants. For the recruitment of participants, convenience sampling with a simplified snowball sampling technique was used. The survey was pre-set to limit to only a single response using their email address. The participants could fill out and submit the completed survey using a computer or cell phone. Social media platforms (WhatsApp, Facebook, Twitter) and email were used for the survey.

Ethical approval was sought from the Institutional Bioethics Committee (IBC), University of Sindh, Jamshoro, Pakistan (Ref. No. ORIC/SU/1134).

### 2.5. Measures

#### 2.5.1. Knowledge

The knowledge section consisted of sixteen close-ended questions regarding the discovery of Mpox, source of information, transmission, symptoms, availability of any treatment or prophylaxis, high-risk groups of patients, case fatality rate, and effectiveness of preventive measures and awareness towards Mpox. For each correct answer, 1 point was assigned, an incorrect answer was given 0 points, and the total knowledge score was measured by adding all points (minimum of 0 and maximum of 16 points). Later, based on the bell curve approach (Mean ± 1SD), knowledge levels were categorized as “Poor”, “Average”, and “Good” for scores less than Mean−1SD, Mean − 1SD to Mean + 1SD, and Mean−1SD, respectively.

#### 2.5.2. Attitude

Four statements were used to measure respondents’ attitudes. The statements were based on attitude towards willingness to get the smallpox vaccine against Mpox, willingness to receive the Mpox vaccine if made available, willingness to pay for vaccine if available, and willingness to receive a free Mpox vaccine offered by the government. Each statement was rated on a 3-point Likert scale, including disagree (1 point), neutral (2 points), and agree (3 points). By adding all the values, a total attitude score was measured. Accordingly, the bell curve approach (Mean ± 1SD) was used to categorize attitude levels as “Negative”, “Neutral”, and “Positive” for scores less than Mean−1SD, Mean−1SD to Mean + 1SD, and Mean−1SD, respectively.

#### 2.5.3. Perception

Respondents’ perceptions were assessed by four statements based on belief in the severity, frequency, concerns regarding Mpox, and its impact on routine activities. Each statement was assigned 3 options, including least likely (1 point), neutral (2 points), and most likely (3 points).

### 2.6. Sample Size Determination

The sample size was determined using the following equation:(1)$$n=\frac{z^{2}pq}{d^{2}}\;n=\frac{1.96^{2}\times 0.5\times \left(1-0.5\right)}{0.05^{2}}\;n=384.16\;\approx \;384$$

The minimum sample size required for our study was 384.

*n* = number of samples

z = 1.96 (95% confidence level)

*p* = prevalence estimate (50% or 0.5)

*q* = (1 − *p*)

*d* = precision limit or proportion of sampling error (0.05)

### 2.7. Statistical Analysis

Data were recorded and analyzed using IBM SPSS software version 27. Continuous variables were presented as means ± standard deviations (SD). In frequencies and percentages, categorical variables were expressed. A comparison of knowledge, attitude, and perception toward Mpox was conducted through Chi-square tests. The Mann–Whitney U test was applied to compare the mean knowledge among different group variables. A univariable and multivariable linear regression was performed to predict the factors associated with knowledge of Mpox. All *p*-values ≤ 0.05 were considered significant.

## 3. Results

### 3.1. Baseline Socio-Demographic Characteristics

As shown in Table 1, a total of 946 respondents participated in the study, out of which the majority were females (514, 54.3%). The mean (SD) age of the respondents was 22.5 (3.5) years, with the majority aged 18–22 years (503, 53.2%). Most of the respondents were from the Sindh region (668, 70.6%), were undergraduates (867, 91.6%), were studying pharmaceutical sciences (669, 70.7%), and had a monthly family income of less than 50,000 PKR (520, 55.0%).

Most of the respondents were not aware of Mpox before 2022 (646, 68.3%), and had not received university-level information on Mpox (748, 79.1%). Most respondents had previously received the COVID-19 vaccine (870, 92.0%) but did not get the flu vaccine (590, 37.6%).

### 3.2. Knowledge of Respondents toward Mpox

Regarding overall knowledge of Mpox, most of the respondents had average knowledge (726, 76.7%), with very few having good knowledge (60, 6.3%). Specifically, most of the respondents were aware that “Mpox infection is associated with typical skin lesions” (766, 81.0%) and “Mpox virus can be prevented by taking standard preventive measures” (592, 62.6%). In addition, very few respondents were aware of “The range of the case-fatality ratio of monkey virus” (214, 22.6%) and that “Asymptomatic patients cannot transmit the Mpox virus to others” (270, 28.5%). However, roughly half of the respondents correctly responded to other knowledge-based questions about Mpox (in the range of 45.5%–59.0%). This is explained in Table 2.

### 3.3. Attitudes of Respondents for Vaccination toward Mpox

Regarding overall attitudes towards Mpox, most respondents had neutral attitudes (648, 68.5%), with the least having negative attitudes (104, 11.0%). Specifically, most of the respondents agreed that “If made available, I am willing to receive Mpox vaccine” (640, 67.7%) and agreed that “I am willing to receive smallpox vaccine to prevent Mpox infection” (530, 56.0%). However, the distribution of respondents on other attitude-based questions towards Mpox was roughly equal for agree, neutral, and disagree (in the range of 28.5%–36.2%), as given in Table 3.

### 3.4. Perception of Respondents toward Mpox

Most of the respondents agreed on “To what extent do you agree that Mpox is a severe infection?” (576, 60.9%), “To what extent are you concerned about Mpox as a public health threat? (570, 60.3%), “To what extent do you agree that Mpox will affect your routine activity in the following year?” (428, 45.2%) and “To what extent do you agree that Mpox is a frequently occurring infection?” (408, 43.1%). However, the distribution of respondents on other attitude-based questions towards Mpox was roughly equal for agree, neutral, and disagree (in the range of 28.5%–36.2%), as shown in Figure 1.

### 3.5. Association between Socio-Demographics with Knowledge and Attitude of Mpox

A significant association was noticed between knowledge of Mpox with the type of academic degree (*p* < 0.001), type of discipline (*p* < 0.001), and region of respondents (*p* < 0.001). In addition, there was a significant association between attitudes toward Mpox with gender (*p* = 0.012), age groups (*p* = 0.002), type of academic degree (*p* < 0.001), and type of discipline (*p* < 0.001), as mentioned in Table 4.

### 3.6. Factors Associated with Knowledge of Mpox Using Linear Regression

Using the univariable analysis, the factors that are associated with knowledge of Mpox were: being a post-graduate (β, −1.65; 95% C, −2.47–0.83; *p* < 0.001), studying pharmaceutical sciences (β, 1.31; 95% CI, 0.82–1.81; *p* < 0.001), studying biological sciences (β, −1.67; 95% CI, −2.58—0.75; *p* < 0.001), studying other disciplines (β, −2.30; 95% CI, −3.12—1.48; *p* < 0.001), aware of Mpox before 2022 (β, 1.64; 95% CI, 1.16–2.13, *p* < 0.001), receiving university-level information on Mpox (β, 2.03; 95% CI, 1.48–2.58; *p* < 0.001), previous COVID-19 vaccination (β, 1.97; 95% CI,1.13–2.80; *p* < 0.001) and previous flu vaccination (β, 0.57; 95% CI, 0.10–1.04; *p* = 0.019).

After adjusting for confounders, using the multivariable regression, the factors associated with knowledge of Mpox were: studying pharmaceutical sciences (β, 2.08; 95% CI, 1.47–2.70; *p* < 0.001), studying medical sciences (β, 1.46; 95% CI, 0.66–2.26; *p <* 0.001), aware of Mpox before 2022 (β, 1.19; 95% CI, 0.71–1.67; *p* < 0.001), receiving university-level information on Mpox (β, 1.56; 95% CI, 1.00–2.12; *p* < 0.001), and previous COVID-19 vaccination (β, 1.46; 95% CI, 0.66–2.25; *p* < 0.001), as explained in Table 5.

## 4. Discussion

Mpox cases continue to rise worldwide; as of 27 November 2022, the cases have surpassed 81,000 globally [6]. However, it is noteworthy that Mpox infection is unknown to most people, physicians, and policymakers in developing countries [22]. For that reason, it is essential to disseminate the “knowledge” about the disease immediately, and the media and scientific community should be used to stabilize the “attitude” of all segments of the community. Finally, proper “practice” to contain the disease would serve the purpose of “Knowledge Attitudes Practices (KAP)” for this severe disease at this volatile moment in time.

The current study included students as study participants from various disciplines, including healthcare as well as non-healthcare students. The presence of a knowledge gap was indicated among the study participants. In general, 76.7% of the study participants had average knowledge, and only 6.3% had good knowledge regarding Mpox disease, its transmission, prevention, and treatment. The present study results align with previous studies revealing defects in knowledge regarding Mpox disease between Italian physicians, Jordanian and Kuwaiti healthcare workers, the general public in Saudi Arabia, Lebanon and Iraq, and medical students from various countries [1,23,24,25]. A study from Saudi Arabia reported that the overall knowledge regarding Mpox disease was poor among the general public and more than half of the respondents had low knowledge about Mpox [26]. A low level of knowledge about Mpox and its prevention measures was also reported in an Italian study [27]. The low level of knowledge is distressing, since public engagement is essential to successfully implement preventive strategies to control and treat possible outbreaks [26]. While comparing the findings with the COVID-19 pandemic infection, it has been reported in a study that people in Sindh, Pakistan, had good knowledge regarding symptoms, route of transmission, preventive measures, and impact of the COVID-19 pandemic [28].

Regarding overall attitudes towards Mpox, most of the respondents had neutral attitudes (68.5%), with the least having negative attitudes (11%). Most of the study participants (56%) agreed to be vaccinated to prevent infection. A survey from the US general population demonstrated that 46% of study participants were willing to get vaccinated against Mpox [29]. Moreover, Salim et al. (2022) reported that 77.3% of internal medicine residents from Indonesia were willing to get vaccinated against Mpox [30]. When comparing the willingness to have COVID-19 vaccination in the general public of Pakistan, it has been reported that 30.3% of the population have a positive attitude towards vaccination [31]. Thus, our study respondents reported much more willingness to get vaccinated. However, it is noteworthy that the attitudes concerning various vaccines, in addition to the level of trust and misinformation, are likely to evolve over time regarding “outcomes and opinions”; therefore, they require frequent evaluation, given their evolving nature.

The key recommendations for preventing Mpox are good hygiene practices, quick identification and isolation of infected individuals, and vaccination [32]. The current study demonstrated that Mpox is a severe infection, which more than half of the participants of the proposed study agreed with (60.9%), and that it is a public health threat (60.3%). It has been stated in another study that “the public health importance of Mpox disease should not be underestimated” [33].

Regarding socio-demographic characteristics of participants associated with the overall knowledge of Mpox disease, the type of academic degree, discipline, and region of respondents were significantly associated with a good level of knowledge. Moreover, there was a significant association between attitudes toward Mpox with gender, age groups, academic degree, and discipline. For example, a study from Iraq reported a better knowledge about Mpox disease among males in the general population [24]. Regarding attitudes, it was reported that attitudes change according to age, level of education, gender, and region of the participants. Further, a study conducted among the general population of Saudi Arabia to assess knowledge regarding Mpox demonstrated that good knowledge was associated with being a healthcare worker, older, employed, having a high income and a higher education, and being married. This is the indication that access to trustworthy information leads to one having high knowledge [26].

At the moment, there is an urgent need to vaccinate people to protect them against Mpox. Therefore, proper strategies are required for vaccinating people. Unequivocally, it is much better to prevent disease in healthy populations than to make an effort to treat a disease in already sick patients. The realignment of vaccination strategies as proposed will work for the common well-being of the human population, particularly for the vulnerable population or those who have close contact with animals such as monkeys or rodents.

Currently, two smallpox vaccines are available in the USA which are equally recommended for Mpox (JYNNEOSTM and ACAM200). These vaccines are administered to people with chance of exposure to any Orthopoxviruses. This is entitled PrEP (pre-exposure prophylaxis). The people who might have a chance to get PrEP include research laboratory workers, clinical laboratory workers, public health workers, and healthcare professionals [34].

Nucleic acid vaccines do not require a complicated manufacturing process as after the immunization, the body becomes a bioreactor of the viral antigen. Thus, the process of vaccine development is cell-free, simpler, and cost- and time-effective. Above all, these types of vaccines are favorably safe. In order to design a potentially universal vaccine that will be effective against Mpox virus, Variola virus (VARV), and Vaccinia virus (VACV), making a multi-epitope vaccine based on the conserved elements of the reasonably selected antigens seems to be an excellent method [35]. Some researchers have developed mathematical or statistical models to evaluate vaccination programs. Bankuru et al. developed a game-theoretic model that might help evaluate different vaccination approaches [36]. However, studies have suggested that timely contact tracing and looped vaccination could limit the virus spread [37]. Our study also emphasized the need for Mpox knowledge improvement among the students who are high-risk representatives of the general population.

Although this study is the first, to our knowledge, to examine KAPs related to Mpox among students of various disciplines in a Pakistani population, it has some limitations. First, the finding of this study may not represent KAPs across Pakistan because of the small sample size. Second, the survey was administered online, including only the participants having access to the internet. Third, the cause-and-effect correlations cannot be distinctly expressed because the current study used a cross-sectional design. Additionally, the study was subject to recall biases and therefore it is important to consider how well the study participants were able to recall their prior knowledge and perceptions and their reporting styles. Most students were studying pharmaceutical sciences, so there was also a selection study bias.

## 5. Conclusions

In conclusion, our study findings indicate that the overall knowledge of Mpox was average in the majority of the university students, with substantial knowledge gaps in most aspects of knowledge. Similarly, most respondents’ overall attitudes towards Mpox were neutral, with the least having negative attitudes. More than half of the respondents (56%) agreed to be vaccinated to prevent infection. Further, the knowledge was significantly associated with the type of academic degree, type of study discipline, and region of respondents. The findings of our study stress the need to promote better knowledge regarding Mpox and its preventive measures. Public awareness programs must be initiated for Mpox to improve adherence to preventive recommendations and may also draw attention to the threats that are of concern for public health because of this zoonotic disease. Our study might be helpful to scientists, laypeople, and policymakers to understand the vaccine and vaccination status against Mpox. However, it is necessary to vaccinate people quickly to fight against this disease.

## Figures and Tables

**Figure 1 vaccines-11-00097-f001:**
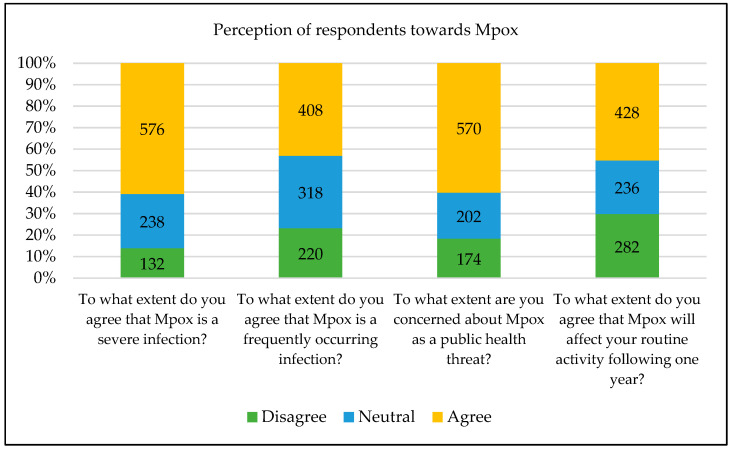
Perception of respondents towards Mpox.

**Table 1 vaccines-11-00097-t001:** Baseline socio-demographic data of the respondents.

Characteristics	Categories	Frequency	Percentage
Gender	Female	514	54.3
	Male	432	45.7
Age (Mean ± SD.)	22.5 ± 3.5		
Age groups (years)	18–22	503	53.2
	23–27	383	40.5
	28 and above	60	6.3
Region	Sindh	668	70.6
	Punjab	166	17.5
	Khyber Pakhtunkhwa	42	4.4
	Azad Kashmir	36	3.8
	Baluchistan	26	2.7
	Gilgit	8	0.8
Education	Undergraduate	867	91.6
	Post-graduate	79	8.4
Discipline	Pharmaceutical Sciences	669	70.7
	Medical Sciences	136	14.4
	Biological Sciences	63	6.7
	Others *	78	8.2
Family income (monthly) in PKR	<50,000 PKR	520	55.0
	50,001–100,000	281	29.7
	>100,000	145	15.3
Aware of Mpox before 2022	No	646	68.3
	Yes	300	31.7
Received university-level information on Mpox	No	748	79.1
	Yes	198	20.9
Previous COVID-19 vaccination	No	76	8.0
	Yes	870	92.0
Previous seasonal flu vaccination	No	590	62.4
	Yes	356	37.6

Keys: * Computer science and information technology, business administration, and commerce, language studies, social sciences; PKR = Pakistani Rupees.

**Table 2 vaccines-11-00097-t002:** General characteristics of respondents.

Variables	Accuracy (*n*)	Frequency
K1. The Mpox virus is not a new discovery.	542	57.3
K2. The Mpox virus circulates only among primates, including humans.	516	54.5
K3. In most cases, Mpox does not present with the symptoms of an uncomplicated influenza-like illness.	430	45.5
K4. Mpox infection is associated with typical skin lesions.	766	81.0
K5. Asymptomatic patients cannot transmit the Mpox virus to others.	270	28.5
K6. European cases of Mpox have been mostly travel-associated.	522	55.2
K7. Currently, there is no specific vaccine against Mpox approved.	550	58.1
K8. Currently, there is no specific drug against Mpox approved.	500	52.9
K9. Recipients of the smallpox vaccine may need further vaccination shots against Mpox.	550	58.1
K10. Mpox causes more severe illness in children than in adults.	548	57.9
K11. Mpox infection is associated with a high rate of systemic complications.	558	59.0
K12. The skin rash associated with Mpox is typically synchronous (in a pattern).	536	56.7
K13. Standard preventive measures are effective in preventing Mpox infection.	592	62.6
K14. Mpox can survive for several days on contaminated surfaces.	544	57.5
K15. Mode of transmission.	546	57.7
K16. The usual case-fatality ratio of Mpox.	214	22.6
Knowledge level		
Poor	160	16.9
Average	726	76.7
Good	60	6.3

Keys: Mpox: Monkeypox

**Table 3 vaccines-11-00097-t003:** Attitudes of respondents toward vaccination toward Mpox.

Statements	Response	Frequency	Percentage
A1. I am willing to receive the smallpox vaccine to prevent Mpox infection.	Disagree	206	21.8
	Neutral	210	22.2
	Agree	530	56.0
A2. If made available, I am willing to receive the Mpox vaccine.	Disagree	148	15.6
	Neutral	158	16.7
	Agree	640	67.7
A3. I am willing to pay to receive a vaccine against Mpox.	Disagree	270	28.5
	Neutral	342	36.2
	Agree	334	35.3
A4. Will you get vaccinated against Mpox (if the government provides a free vaccine)?	Disagree	270	28.5
	Neutral	342	36.2
	Agree	334	35.3
Attitude levels	Negative	104	11.0
	Neutral	648	68.5
	Positive	194	20.5

**Table 4 vaccines-11-00097-t004:** Association between socio-demographics with knowledge and attitudes regarding Mpox.

Variable	*n* (%)	Knowledge			*p*-Value	Attitude			*p*-Value
		Poor	Average	Good		Negative	Neutral	Positive	
Gender									
Female	514 (54.3)	88	400	26	0.210	48	344	122	0.012
Male	432 (45.7)	72	326	34		56	304	72	
Age groups									
18–22	503 (53.2)	86	385	32	0.184	62	361	80	0.002
23–27	383 (40.5)	60	295	28		34	247	102	
28 and above	60 (6.3)	14	46	0		8	40	12	
Academic									
Undergraduate	867 (91.6)	134	675	58	<0.001	96	606	165	<0.001
Post-graduate	79 (8.4)	26	51	2		8	42	29	
Discipline									
Pharmaceutical Sciences	669 (70.7)	92	529	48	<0.001	60	471	138	<0.001
Medical Sciences	136 (14.4)	16	112	8		22	80	34	
Biological Sciences	63 (6.7)	22	41	0		6	41	16	
Others *	78 (8.2)	30	44	4		16	56	6	
Region									
Sindh	668 (70.6)	98	524	46	<0.001	74	464	130	0.057
Punjab	166 (17.5)	42	110	14		12	116	38	
Khyber Pakhtunkha	42 (4.4)	2	40	0		8	22	12	
Azad Kashmir	36 (3.8)	12	14	0		6	18	2	
Baluchistan	26 (2.7)	2	6	0		2	4	2	
Gilgit	8 (0.8)	4	32	0		2	24	10	

Keys: * Computer science and information technology, business administration, and commerce, language studies, social sciences; PKR = Pakistani Rupees.

**Table 5 vaccines-11-00097-t005:** Factors associated with knowledge of Mpox using linear regression.

Variables	Univariable Analysis			Multivariable Analysis		
	Beta Coeff	95% CI	*p*-Value	Beta Coeff	95% CI	*p*-Value
Age	0.013	−0.05–0.08	0.697	-	-	-
Gender						
Male	0.28	0.24—0.19	0.241	-	-	-
Female	1			-	-	-
Academic						
Post-graduate	−1.65	−2.47–0.83	<0.0001	-	-	-
Undergraduate	1			-	-	-
Discipline						
Pharmaceutical Sciences	1.31	0.82–1.81	<0.0001	2.08	1.47–2.70	<0.0001
Medical Sciences	0.05	−0.61–0.70	0.889	1.46	0.66–2.26	<0.0001
Biological Sciences	−1.67	−2.58–0.75	<0.0001	-	-	-
Others *	−2.30	−3.12–1.48	<0.0001	-	-	-
Income (PKR)						
<50,000	−0.11	−0.57–0.36	0.655	-	-	-
50,001–100,000	0.38	−0.12–0.88	0.138	-	-	-
>100,000	−0.41	−1.05–0.23	0.206	-	-	-
Aware of Mpox before 2022						
Yes	1.64	1.16–2.13	<0.0001	1.19	0.71–1.67	<0.0001
No	1			1		
Received University-level inform						
Yes	2.03	1.48–2.58	<0.0001	1.56	1.00–2.12	<0.0001
No	1			1		
Previous COVID-19 vaccination						
Yes	1.97	1.13–2.80	<0.0001	1.46	0.66–2.25	<0.0001
No	1			1		
Previous seasonal flu vaccination						
No	0.57	0.10–1.04	0.019	-	-	-
Yes	1					

Keys: * Computer science and information technology, business administration, and commerce, language studies, social sciences; PKR = Pakistani Rupees.

## Data Availability

Data will be available upon request.

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
