# Peer review of "Monkeypox Cross-Sectional Survey of Knowledge, Attitudes, Practices, and Willingness to Vaccinate among University Students in Pakistan"

_vaccines, 2022, doi:10.3390/vaccines11010097_

Round 1

Reviewer 1 Report

Thanks for the opportunity to review this interesting manuscript.

In the current study, Narendar Kumar et al. investigated a timely topic, namely the knowledge, attitude and practices of University students in Pakistan towards monkeypox. The rapid increase in the number of monkeypox cases and the declaration by the WHO that this re-emerging infection is a public health emergency of international concern with more than 80,000 cases in the year 2022 alone shows the importance and timeliness of this study.

In addition, the authors investigated the willingness to get vaccinated in case monkeypox vaccination becomes a necessity to control cases.

The major results pointed to an average level of monkeypox knowledge among the study participants, variable levels of willingness to get vaccinated against monkeypox and relatively high level of concern regarding monkeypox severity and public health threats.

Specific comments that hopefully can help the authors to improve the quality of the final version of the manuscript:

1. In the Abstract, please try to rephrase a few sentences to make the flow of text more concise. E.g., there is no need to mention that the study was conducted according to STROBE guidelines or that SPSS was used for statistical analysis.

2. In the Abstract, please delete “the effect of” in line 20.

3. In the Abstract, please replace the word “do” by the word “did” in line 27.

4. In the Abstract, and throughout the manuscript, the authors can benefit from using the updated term for monkeypox "mpox" to reduce stigma and other issues associated with prior terminology as advocated by the WHO. Source: https://www.who.int/news/item/28-11-2022-who-recommends-new-name-for-monkeypox-disease

5. In the Abstract, the authors are recommended to highlight the results of willingness to get mpox since this is a major finding of the study.

6. In the Introduction, the authors can benefit from the recent reviews published on mpox, to highlight the major topics that should be introduced to the reader including mpox vaccination and the role of public health measures in the preparedness to respond to the outbreak including the improvement of knowledge levels among university students and health professionals. A suggested reference: https://www.mdpi.com/1999-4915/14/10/2155

7. In the Methods, please check the references from which the research instrument was adopted since the current references (12-15) were addressing COVID-19 rather than monkeypox. Suggested references on the same instrument include:

https://www.mdpi.com/2076-393X/10/12/2022

https://www.mdpi.com/2414-6366/7/12/411

https://www.tandfonline.com/doi/full/10.1080/20477724.2020.1743037

8. Please specify the outlets used for survey distribution (e-mails only, personal communication, social media platforms, etc.)

9. An important limitation that should be highlighted by the authors is the severe selection bias as a majority of participants belonged to Pharmaceutical Sciences discipline with subsequent possible effect of higher knowledge and more positive attitude towards medically related topics.

Thank you!

Author Response

Point 1: 1. In the Abstract, please try to rephrase a few sentences to make the flow of text more concise. E.g., there is no need to mention that the study was conducted according to STROBE guidelines or that SPSS was used for statistical analysis.

Response 1: Thank you so much for your kind suggestion reviewer. The Abstract has been rephrased. STROBE guidelines or that SPSS was used for statistical analysis, are removed from the abstract.

Point 2: In the Abstract, please delete “the effect of” in line 20.

Response 2: Thank you so much for your kind suggestion reviewer. The effect is deleted from the abstract.

Point 3: In the Abstract, please replace the word “do” with the word “did” in line 27.

Response 3: Thankyou dear reviewer for pointing out, the word do is replaced by did.

Point 4: In the Abstract, and throughout the manuscript, the authors can benefit from using the updated term for monkeypox "Mpox" to reduce stigma and other issues associated with prior terminology as advocated by the WHO. Source: https://www.who.int/news/item/28-11-2022-who-recommends-new-name-for-monkeypox-disease

Response 4: Thank you for the kind suggestion. Throughout the manuscript, the word monkeypox "Mpox" is replaced.

Point 5. In the Abstract, the authors are recommended to highlight the results of willingness to get mpox since this is a major finding of the study.

Response 5: Thank you for the kind suggestion. The results of willingness to get Mpox is added in the abstract.

Point 6. In the Introduction, the authors can benefit from the recent reviews published on mpox, to highlight the major topics that should be introduced to the reader including mpox vaccination and the role of public health measures in the preparedness to respond to the outbreak including the improvement of knowledge levels among university students and health professionals. A suggested reference: https://www.mdpi.com/1999-4915/14/10/2155

Response 6: Thank you for the kind suggestion. This review is very interesting and was a great help in terms of the role of public health measures in the preparedness to respond to the outbreak including the improvement of knowledge levels among university students and health professionals. I have added it to my study.

Point 7. In the Methods, please check the references from which the research instrument was adopted since the current references (12-15) were addressing COVID-19 rather than monkeypox. Suggested references on the same instrument include:

https://www.mdpi.com/2076-393X/10/12/2022

https://www.mdpi.com/2414-6366/7/12/411

https://www.tandfonline.com/doi/full/10.1080/20477724.2020.1743037

Response 7: Thank you for the kind suggestion dear reviewer. I have changed the references according to Mpox studies.

Point 8. Please specify the outlets used for survey distribution (e-mails only, personal communication, social media platforms, etc.)

Response 8: Dear reviewer, social media platforms (WhatsApp, Facebook, Twitter) and email were used for the survey. The modes are added to the manuscript and highlighted.

Point 9. An important limitation that should be highlighted by the authors is the severe selection bias as a majority of participants belonged to Pharmaceutical Sciences discipline with subsequent possible effect of higher knowledge and more positive attitude towards medically related topics.

Response 9: Thank you for the kind suggestion, the selection bias towards the majority of pharmaceutical students has been added to the limitations.

Reviewer 2 Report

The authors are presenting a survey among a general university student population sample in Pakistan, on knowledge on mpox, but also vaccination intentions.

I would like to share the following comments with the authors:

I do not understand the need for this research. During the wave of the global outbreak no cases were diagnosed in Pakistan (Najeeb et al 2022), and also WHO does not report any cases along the MSM transmission routes reported in other countries. 

Even if one would expect future outbreaks in Pakistan, a general student population would be the last one affected, most likely. The chosen sample is simply not targeting the right population.

Even from a (future) healthcare provider information perspective the sampling is off, then no non-medical students should have been included. By the same token, we do not need a survey to demonstrate that there is low knowledge in this population, this simply requires educating future healthcare providers. To show an urgent gap, a survey among practicing healthcare providers would have been more applicable.

There are papers available on the Pakistani context and should be cited, such as Mansoor et al, 2022 in Annals of Medicine and Surgery, or Najeeb et al, 2022.

I find the focus on vaccination misguided as well, no other country would have offered a mpox vaccination to this population. Thus assessing their vaccination readiness is not relevant and potentially even harmful as it can elicit fears.

No information about ethics approval is provided.

Minor points:

The new and current abbreviation mpox should be used

Author Response

Point 1: I do not understand the need for this research. During the wave of the global outbreak no cases were diagnosed in Pakistan (Najeeb et al 2022), and also WHO does not report any cases along the MSM transmission routes reported in other countries.

Response 1: Thank you for pointing this out dear reviewer, As of May 23, 2022, Pakistan has confronted two rarely occurring cases of the zoonotic monkeypox infection that has spread around the globe. According to the doctors of Lahore Services Hospital, two cases of monkeypox were detected in Lahore Jinnah Hospital, Pakistan. The patients were isolated and properly treated in separate wards. After the detection of these cases, the National Institute of Health strongly warned the healthcare settings of the country to deal carefully with the disease (https://tribune.com.pk/story/2357654/who-calls-emergency-meeting-as-monkeypox-cases-top-100-in-europe). Considering the burden that has been inflicted by the COVID-19 pandemic on Pakistan's already struggling healthcare system, several precautionary measures need to be ensured to prevent its continuous decimation. In the past, co-epidemics and co-occurrences of viral diseases such as dengue fever, zika, chikungunya, Crimean Congo hemorrhagic fever, measles, and poliomyelitis along with the COVID-19 infection have resulted in numerous casualties that could've been prevented by taking the advanced precautionary measures. The struggling healthcare system will be on the verge of collapse if monkeypox starts to spread. Pakistan does not have any diagnostic facility for the virus, the health department has declared samples can be sent abroad for testing in case of emergencies which further threatens the spread. To tackle this situation, there is a need to have adequate knowledge regarding the presenting signs and symptoms of the disease to assure the timely quarantine of suspected patients instead of symptomatic treatments only. Additionally, hospitals should be prepared with well-equipped isolation units to quarantine patients immediately to limit the spread of the contagious virus.

Point 2. Even if one would expect future outbreaks in Pakistan, a general student population would be the last one affected, most likely. The chosen sample is simply not targeting the right population. Even from a (future) healthcare provider information perspective the sampling is off, then no non-medical students should have been included. By the same token, we do not need a survey to demonstrate that there is low knowledge in this population, this simply requires educating future healthcare providers. To show an urgent gap, a survey among practicing healthcare providers would have been more applicable.

Response 2: Dear reviewer, the total literacy rate of Pakistan is 32.33%. The number of students who get a university education is less than 30%. Due to unawareness and illiteracy, there is a lack of basic understanding of basic health rights. So, if any pandemic strikes the university students should be capable enough to spread awareness among the masses. This is the main reason to target the population i.e., university students. The main aim was to explore knowledge, attitude, perceptions, and willingness to vaccination among students studying in Universities in Pakistan. Most of the students included in the study were from a medical background. Pakistan is a low-middle-income country, with inequitable distribution of scarce resources. In the year 2021, the government spent 1.2 percent of the GDP on health, this amount is far less than the WHO recommendation of Five percent. The responsiveness of the health system is another major issue, added on by a reactive instead of a proactive approach, we usually identify problems when they have already been complicated. The same holds true when we are faced with disasters of varying intensity.

Point 4: There are papers available on the Pakistani context and should be cited, such as Mansoor et al, 2022 in Annals of Medicine and Surgery, or Najeeb et al, 2022.

Response 4: Thank you for the kind suggestion dear reviewer. Najeeb et al, 2022, are cited in the manuscript.

Point 5. I find the focus on vaccination misguided as well, no other country would have offered a mpox vaccination to this population. Thus, assessing their vaccination readiness is not relevant and potentially even harmful as it can elicit fears.

Response 5: Dear reviewer, Vaccine hesitancy remains a substantial challenge for Pakistan amid various conspiracy theories. The failure to eradicate polio from the country is primarily attributed to such theories. Of these, alleged poor quality of vaccines, questioning of dosing recommendations, religious prohibitions (“infidel vaccine”), and rumors related to the presence of active virus in the vaccines are some leading claims obstructing the anti-polio campaign in the country. Unfortunately, the same was the case with the COVID-19 vaccine in Pakistan. In the country, where vaccine hesitancy is a prime barrier to curbing vaccine-preventable diseases, such conspiracy narratives may plant seeds of resistance against upcoming vaccination programs. So, the willingness is observed to check the response of the educated class of the population.

Point 6. No information about ethics approval is provided.

Response 6: Ethical approval was sought from the Institutional Bioethics Committee (IBC), University of Sindh, Jamshoro, Pakistan (Ref. No. ORIC/SU/1134). It is mentioned and highlighted in the manuscript.

Point 7. The new and current abbreviation mpox should be used

Response 7: Dear reviewer, thank you so much for your kind suggestion. The current abbreviation Mpox” is used throughout the manuscript.

Reviewer 3 Report

- I would like to congratulate the authors on conducting their research in a timely manner to explore the monkeypox perceptions among university students 

- I suggest to use the new WHO name for monkeypox: mpox 

- Introduction and discussions could be further empowered:

 The background comparison of mpox in several countries could be further supported by literature, comparing the HCW, public and university students, such as:

https://doi.org/10.3390/vaccines10122071

https://doi.org/10.3390/vaccines10091408

doi: 10.2807/1560-7917.ES.2022.27.40.2200734

https://doi.org/10.1016/j.tmaid.2022.102426

- also in the introduction, please add to Ref 6 another statement that “In the 2022 mpox, the fever is often absent or occurs after the appearance of the rash. Also, the rash presents mainly in the anogenital region and face before disseminating in the body, with lesions showing regional pleomorphism.” doi: 10.1186/s12879-022-07900-7.

- Methods: is English the official language for all universities in Pakistan?

- Results:

- can you estimate the response rate for your survey?

- the responses came mainly from the pharmacy students, this has to be highlighted in the abstract and limitations section 

- I suggest switching part of the many tables into one or two figures, if possible, to make it more eye- catching for the readers 

- the Title is all CAPITALIZED, is that the Journal style?

Author Response

Point 1: - I suggest to use the new WHO name for monkeypox: mpox

Response 1: Thank you for pointing this out dear reviewer, the new term by WHO, Mpox is used in the manuscript.

Point 2. - Introduction and discussions could be further empowered:

The background comparison of mpox in several countries could be further supported by literature, comparing the HCW, public and university students, such as:

https://doi.org/10.3390/vaccines10122071

https://doi.org/10.3390/vaccines10091408

doi: 10.2807/1560-7917.ES.2022.27.40.2200734

https://doi.org/10.1016/j.tmaid.2022.102426

Response 2: Dear reviewer, thank you so much for your kind suggestion. The following articles were a great help to empower Introduction and discussion. The changes are done and highlighted.

Point 3: Is English the official language for all universities in Pakistan?

Response 3: Yes. English is the instruction medium in all the public universities of Pakistan.

Point 4. Can you estimate the response rate for your survey?

Response 4: Dear Reviewer, the response rate could not be estimated as the survey was conducted using an online link, and forwarded to many people using the snowball method. However, a total of 972 responses were recorded. Of the 946 who completed the survey, and they were included in the study.

Point 5.  the responses came mainly from the pharmacy students, this has to be highlighted in the abstract and limitations section 

Response 5:  Dear Reviewer, thank you for your suggestion, it has been incorporated in the abstract and limitations and highlighted.

Point 6. I suggest switching part of the many tables into one or two figures, if possible, to make it more eye- catching for the readers

Response 6: Dear reviewer, thank you for the kind suggestion. A table “Perception towards Mpox response” has been converted to the figure.

Point 7.  the Title is all CAPITALIZED, is that the Journal style?

Response 7:  The title is changed according to the journal style.

Round 2

Reviewer 1 Report

Thanks for properly addressing all my previous concerns

Best wishes

Author Response

Dear Reviewer,

Thank you for your kindness and response.

Reviewer 2 Report

I have read a previous version of the manuscript. The authors respond well to my concerns about the need of the current paper in their reply to the editor, I would strongly suggest to include these arguments in the paper as well. In particular, there should be more information about the precarious and fragile healthcare system in Pakistan and the healthcare literacy of the population (both backed with references). Then the reader is able to understand the need for the paper better.

My other suggestions have been adequately dealt with.

Author Response

Dear Reviewer,

Thank you for your kind suggestions. More information about the precarious and fragile healthcare system in Pakistan and the healthcare literacy of the population (both backed with references) is added to the revised manuscript.

Thanks.